# Equivalent Method of Joint Interface Based on Persson Contact Theory: Virtual Material Method

**DOI:** 10.3390/ma12193150

**Published:** 2019-09-26

**Authors:** Renxiu Han, Guoxi Li, Jingzhong Gong, Meng Zhang, Kai Zhang

**Affiliations:** 1College of Mechatronics Engineering and Automation, National University of Defense Technology, No.109 Deya Street, Changsha 410073, China; lgx2020@sina.com (G.L.); zhangmengchn@126.com (M.Z.); kai277063148@163.com (K.Z.); 2School of Mechanical Engineering, Hunan International Economics University, No.822 Fenglin Street, Changsha 410205, China; jzgong@nudt.edu.cn

**Keywords:** equivalent method of interface, Persson contact theory, virtual material method

## Abstract

An accurate equivalent method of metal joint interface is of great significance to optimize the dynamic performance of the whole machine. Therefore, it is necessary to establish an accurate equivalent method of joint interface. The virtual material method is a precise equivalent method of joint interface. The traditional virtual material method is based on the M–B fractal contact theory. By modeling the contact mechanics of the joint interface, the physical properties of the virtual material are obtained separately, such as elastic modulus, Poisson’s ratio and density. In this paper, Persson contact theory is used to establish the interface contact mechanics model to find the physical properties of virtual materials. The virtual material methods constructed by two theories are respectively applied to the modal simulation to obtain the natural frequencies of the joint interface. By comparing the natural frequencies obtained by modal experiment and modal simulation, it is found that the natural frequencies obtained by the virtual material method based on Persson contact theory are closer to the results obtained by the modal experiment, and the error is within 5%. The error of the natural frequencies obtained by the virtual material method based on the M–B fractal contact theory is within 10%. Therefore, the Persson contact theory can establish a more accurate equivalent method of metal’s joint interface.

## 1. Introduction

The part connecting the upper and lower surfaces between the components is called the mechanical joint interface, which is ubiquitous in the mechanical structure. Through the study of the joint interface, the scholars found that the 50% total stiffness, and more than 90% total damping of the machine tool comes from the joint interface [1]. Therefore, the establishment of an accurate equivalent method of joint interface is of great significance for engineering applications. According to the recent studies [2,3,4,5,6], the equivalent method of joint interface includes a virtual material method, spring damping method, and finite element method. By comparing the mode shapes and the natural frequencies obtained by modal experiment and modal simulation [5], it is found that the virtual material method can better simulate the metal joint interface than the other two methods.

A large number of studies have shown that [7,8,9,10] most of the machined surfaces have statistical self-affine characteristics. The rough surface similarity is unique at different scales. Even at the atomic scale, the surface morphology still has multi-scale, self-affine and non-stationary features, namely fractal characteristics. The function simulates a rough surface with good agreement with the measured surface. Its fractal roughness parameter G, fractal dimension D and Hurst exponent H are dimension independent. Therefore, the fractal theory can reflect the actual surface contour of the assembly joint interface more realistically and reliably. In addition, many scholars [11,12,13,14] study the relevant characteristics of the joint interface through fractal theory, and explain the importance of fractal theory to the research of the joint interface.

The virtual material method was proposed by Tian Hongliang [2]. This method uses the M–B fractal contact theory [15,16] to establish the contact mechanics model of the mechanical joint interface, and obtains the physical parameters of the virtual material from the microscopic point of view, such as Poisson’s ratio, density, elastic modulus, etc. Through the comparison of experiments and simulations [2,5], it is found that the error of the virtual material method established by the M–B fractal contact theory is large, which is about 10%. However, the M–B contact mechanics model has certain limitations. The contact mechanics model ignores the interaction between the asperities. When the pressing force is small, this assumption is approximately true, but when the pressing force is relatively large, the interaction between the asperities cannot be ignored. Therefore, it will affect the establishment of the equivalent model. In recent years, Persson et al. used mathematical methods such as fractal geometry and frequency domain transformation to draw a set of contact mechanics theory considering scale effect [17,18,19,20,21,22,23,24,25,26,27]. The mechanics model considers the effect of scale effects on contact behavior and the interaction between asperities. In order to consider the effect of the surface roughness scale effect on the contact behavior, the model assumes that the deformation of the surface asperities or dimples is approximately equal to the height of the surface asperities or the depth of the dimples. This assumption applies only to situations where the pressing force (actual contact area) is large, but it does not apply to situations where the pressing force (actual contact area) is relatively small. This boundary condition is suitable for the metal joint interface studied in this paper. Through experimental and theoretical calculations [26], it is found that, with the increase in pressure, the theoretically calculated stiffness between the joint interface can be well fitted to the actual stiffness between the joint interface.

Constructing an accurate equivalent method of joint interface can accurately analyze the dynamic performance of the whole machine. In this paper, two different joint interface contact mechanics theories are used to construct the equivalent model of the joint interface, namely the virtual material method. Through the comparative analysis of modal experiment and modal simulation, a better theoretical method for constructing the equivalent model of the joint interface is obtained.

## 2. Materials and Methods 

The specimens, made of the same material (GB: 45#, JIS: S45C, ANSI: 1045, DIN:C45), had L = 0.2 m length, l = 0.06 m thickness. The specimens were pressed by the pressure 0.125 MPa by six bolts, as shown in Figure 1. The specimens are machined by milling and have a surface roughness of approximately 3.2 μm. The physical properties of the specimens are shown in Table 1. Considering that the joint interface of the machine is usually composed of the same material, the specimens of the same material are selected. 

The microscopic contact portion of the two contact surfaces of the fixed joint interface is assumed to be a virtual isotropic material as described in Ref. [2]. The length and width of the virtual material are the length and width of the specimens, respectively. The elastic modulus, Poisson’s ratio, and density of the virtual material are obtained by some theoretical methods and the thickness of the virtual material is determined according to the actual gap between the two surfaces, which is the virtual material method. In the following, the virtual materials are constructed by Persson contact theory and M–B fractal contact theory, respectively, so as to find a better theoretical method for constructing the equivalent model of joint interface.

### 2.1. Virtual Material Method Based on Persson Contact Theory

#### 2.1.1. The Surface Roughness Power Spectrum

The data of specimens’ surfaces height are acquired by a coordinate measuring machine (ACCURA II AKTIV; Carl Zeiss AG, Oberkochen, Germany, as shown in Figure 2a. The sampling interval of the data is a=0.001 m. Experiment takes 200×200 data points of surface height. The data are fitted to the surface of the specimens by software (MATLAB 2014; MathWorks, Inc., Natick, MA, USA), as shown in Figure 2b.

According to the authors in [27], the surface roughness power spectrum C(q) can be written as:(1)$$C\left(q\right)=\frac{{\left(2\pi \right)}^{2}}{A}\langle {\left|h_{A}\left(q\right)\right|}^{2}\rangle ,$$
where A is the surface area, $h_{A}\left(q\right)$ is the Fourier transform of the surface height h(x), which can be written as:(2)$$h_{A}\left(q\right)\approx \frac{a^{2}}{{\left(2\pi \right)}^{2}}\sum_{n}h_{n}\left(X_{n}\right)e^{-i\left(2\pi m_{x}n_{x}a+2\pi m_{y}n_{y}a\right)/L},$$
where *a* is the length of the specimen, and $n_{x}=1,2,\cdots ,200$, $n_{y}=1,2,\cdots ,200$, $m_{x}$ and $m_{y}$ are integers between 0–199.

Mean square roughness amplitude can be obtained from the acquired data of the surface height:(3)$$h_{rms}^{2}=\langle h^{2}\rangle =\frac{1}{N^{2}}{\sum_{n}\left(h_{n}-\bar{h}\right)}^{2},$$
where N=200, $\bar{h}$ is the average of the surface height, 〈⋯〉 indicates statistical average of the physical quantities in angle brackets.

Substituting the values of the surface profile height into Equations (1) and (2), the surface roughness power spectrum can be obtained, as shown in Figure 3, where $q_{r}=2\pi /L$ is characteristic frequency, $q_{0}=2\pi /\lambda_{0}$ is minimum cut-off frequency, $q_{1}=2\pi /\lambda_{1}$ is maximum cut-off frequency, $\lambda_{0}$ is long-wavelength roll-off and $\lambda_{1}$ is short wavelength cut-off.

#### 2.1.2. The Normal Stiffness Model of Interface

According to the authors in [26], the dimensionless elastic energy stored in the Hertz mesoscale deformation field for depth of indentation is written as: (4)$$U_{0}^{\prime }=\frac{4}{3}\kappa_{0}\chi^{-1/\left(1+H\right)}{\left(\frac{3F^{\prime }}{4}\right)}^{\left(1+2H\right)/\left(1+H\right)},$$
where $\kappa_{0}=2/5$ is the root-mean-square curvature of the surface, $F^{\prime }$ is the dimensionless normal force, H=3-D is the Hurst exponent, and χ is the dimensionless prefactor which is obtained by:(5)$$\chi ={\left(\frac{2-H}{Hs_{0}}\right)}^{1/2}\frac{\pi^{H-2}}{\beta q_{0}h_{rms}},$$
where $\beta =\sqrt{8/3\pi }$ when $q_{1}/q_{r}\gg 1$, ${1/s}_{0}=1+H\left[1-{\left(q_{r}/q_{0}\right)}^{2}\right]$.

The dimensionless elastic deformation energy [25] that is stored in microasperity contacts within the Hertz mesoasperity contact region is obtained by:(6)$$U_{1}^{\prime }=\frac{4}{3}\kappa_{1}\chi^{-1/\left(1+H\right)}{\left(\frac{3F^{\prime }}{4}\right)}^{\left(1+2H\right)/\left(1+H\right)},$$
where $\kappa_{1}=\gamma {\left(\frac{2-H}{\pi^{4}\beta^{2}H}\right)}^{1/2}$ is the prefactor, γ≈0.4 according to Ref. [26].

The total dimensionless elastic energy is now given by the sum of the two contributions Equations (4) and (6):(7)$$U^{\prime }=\frac{4}{3}\kappa \chi^{-1/\left(1+H\right)}{\left(\frac{3F^{\prime }}{4}\right)}^{\left(1+2H\right)/\left(1+H\right)},$$
(8)$$\kappa =\kappa_{0}+\kappa_{1}.$$
The total dimensionless stiffness of the interface is written as:(9)$$k^{\prime }=\frac{F^{\prime }}{dU^{\prime }/dF^{\prime }}=\theta {\left(\frac{F^{\prime }}{q_{0}h_{rms}s^{1/2}}\right)}^{1/\left(1+H\right)}.$$

Dimensionless to Equation (9) is written as:(10)$$\frac{Kh_{rms}}{E^{*}}=\theta {\left(\frac{h_{rms}}{2\pi \lambda_{r}}\frac{\lambda_{r}^{2}}{L^{2}}\right)}^{H/\left(1+H\right)}{\left(\frac{p}{s^{1/2}E^{*}}\right)}^{1/\left(1+H\right)},$$
where θ is the prefactor that is obtained by Ref. [26], $p=F/A_{0}$ is the surface pressure, and $E^{*}$ is the equivalent elastic modulus.

The elastic modulus E and the shear modulus $G_{\tau }$ of the interface can be written as [28]:(11)$$E=hKn/A_{0},$$
(12)$$G_{\tau }=K_{\tau }h/A_{0},$$
where $K_{\tau }$ is the tangential contact stiffness, and h=0.8 is the thickness of the virtual material according to Ref. [2]. The ratio of the tangential contact stiffness to the normal contact stiffness is 0.25–0.35. Without loss of generality, $K_{\tau }/K_{n}$ is taken as 0.35, according to Ref. [29].

The Poisson’s ratio and density of the virtual material can be described as [4,30] respectively:(13)$$\upsilon =\frac{E}{2G_{\tau }}-1,$$
(14)$$\rho =\frac{\rho_{1}h_{1}+\rho_{2}h_{2}}{h_{1}+h_{2}},$$
where $\rho_{1}$, $\rho_{2}$ and $h_{1}$, $h_{2}$ are the density and thickness of specimen1 and specimen2, respectively.

The physical properties of the virtual material obtained by Equations (10)–(14) are as follows: $E_{1}\;=\;0.0591\;\mathrm{Gpa}$, $\nu_{1}=0.429$, $\rho_{1}=7850\;\mathrm{kg}/m^{3}$ and $h_{1}=0.8\;\mathrm{mm}$.

### 2.2. Virtual Material Method Based on M–B Fractal Contact Theory

The shape of the rough microcontacts on the actual surface is usually an ellipsoid. Since the contact area of the ellipsoid is much smaller than the radius of the curvature of itself, the microcontact can be approximated as a sphere. The contact between the two planes can be seen as a series of bumps in contact with each other. The contact of the two sphere microcontacts is shown in Figure 4.

According to the authors in [31], the normal load of an elastic microcontact is obtained from the hemispherical positive stress of hemisphere:(15)$$F=\frac{4}{3}E^{*}\sqrt{R_{0}}d^{\frac{3}{2}},$$
where $R_{0}$, d, $E^{*}$ are the equivalent curvature radius of the two contacting microcontacts, the deformation of contact point as shown in Figure 4, and the equivalent elastic modulus of the two contact rough surfaces, respectively:(16)$$R_{0}=\frac{R_{1}\times R_{2}}{R_{1}+R_{2}},$$
(17)$$d=G^{D-1}{a^{\prime }}^{1-0.5D},$$
(18)$$\frac{1}{E^{*}}=\frac{1-\mu_{1}^{2}}{E_{1}}+\frac{1-\mu_{2}^{2}}{E_{2}},$$
where $\mu_{1}$, $\mu_{2}$ and $E_{1}$, $E_{2}$ are Poisson’s ratio and elastic modulus of two objects that constitute the joint interface; $a=a^{\prime }/2=\pi R_{0}d$ is the area of a micro contact point. G and D are fractal parameters. According to the authors in [2], fractal parameters are determined by the roughness of the surface of the joint interface. When the metal surface roughness ranges from 0.4 μm to 3.2 μm, the values of G and D are more accurate.

From Equation (15), the actual normal contact compressive stress of an elastic microcontact is written as: (19)$$P=\frac{F}{\pi r^{2}}=\frac{F}{a}=\frac{4E^{*}}{3\pi }\sqrt{\frac{d}{R_{0}}}=\frac{4E^{*}\sqrt{\varepsilon }}{3\pi },$$
where $\varepsilon =d/R_{0}$ is the normal compressive strain of an elastic micro-contact point.

Differentiating Equation (19), the elastic modulus of two contacting microcontacts is expressed as:(20)$$e=\frac{dP}{d\varepsilon }=\sqrt{\frac{2}{9\pi^{3}}}E^{*}G^{1-D}{a^{\prime }}^{0.5D-0.5}.$$

The statistical distribution of the truncated micro-contact area $a^{\prime }$ can be described as [31]:(21)$$n\left(a^{\prime }\right)=0.5D\psi^{1-D/2}{a_{1}^{\prime }}^{0.5D}{a^{\prime }}^{-1-0.5D}\begin{array}{cc} & \left(0<a^{\prime }\leq a_{1}^{\prime }\right) \end{array},$$
where ψ describes the domain extension factor for the micro-contact size distribution associated with D:(22)$$a_{1}^{\prime }=2a_{1}=\frac{2\left(2-D\right)}{D}\psi^{0.5D-1}A_{r},$$
(23)$$a_{2}^{\prime }=2a_{2}=2G^{2}{\left(\frac{2E^{*}}{H}\right)}^{\frac{2}{D-1}},$$
where $a_{1}$ is the truncated area of the largest elastic micro-contact, $a_{2}$ is the critical truncated area demarcating the elastic and plastic deformation regimes, $A_{r}$ is actual contact area and H is the hardness of material.

The equivalent elastic modulus can be obtained as:(24)$$E=\frac{\int_{{a^{\prime }}_{2}}^{{a^{\prime }}_{1}}en\left(a^{\prime }\right)ada^{\prime }}{A_{0}}=\frac{2^{\left(2D-9\right)/2}E^{*}D\psi^{1-D/2}G^{1-D}{a^{\prime }}^{D/2}\left({a_{1}^{\prime }}^{0.5}-{a_{2}^{\prime }}^{0.5}\right)}{3\pi^{\left(1+D\right)/2}{\left(\ln\gamma \right)}^{1/2}A_{0}}\begin{array}{cc} & \left(a_{1}^{\prime }<a^{\prime }<a_{2}^{\prime }\right) \end{array},$$
where γ is the scaling parameter, and $A_{0}$ is nominal contact area.

The physical properties of the virtual material obtained by Equations (11)–(24) are as follows: $E_{2}=0.1253\;\mathrm{Gpa}$, $\nu_{2}=0.429$, $\rho_{2}=7850\;\mathrm{kg}/m^{3}$ and $h_{2}=0.8\;\mathrm{mm}$.

## 3. Modal Simulation, Modal Experiment and Results

### 3.1. Modal Simulation

The virtual material method is modeled in three dimensions through software (SOLIDWORKS 2018; Dassault Systemes, Waltham, MA, USA). The model is imported into software (ANSYS Workbench 14.0; ANSYS, Inc., Pittsburgh, PA, USA). The physical parameters listed in Table 1 are assigned to specimen1 and specimen2. The physical parameters obtained by Persson contact theory and M–B fractal contact theory are respectively assigned to virtual materials. The mesh diagram of the model is shown in Figure 5. The grid has a total of 140,517 nodes and 75,588 elements. The pressing force of 0.125MPa is applied to the specimens as the boundary condition. Finally, the modal analysis is performed to obtain the first five natural frequencies of the specimens, as shown in Table 2.

### 3.2. Modal Experiment

The experimental setup (PSV-I-500; Polytec GmbH, Waldbronn, Germany) is as shown in Figure 6a. The experimental specimens are suspended by two elastic ropes in front of the exciter to reach the boundaryless condition and excited by an exciter. Experimental data are acquired by laser scanning specimens. Finally, the processing of the data is shown in Figure 6b. The first five natural frequencies of the specimens are shown in Table 3.

### 3.3. Results

The first five natural frequencies and errors obtained from modal experiment and modal simulation are shown in Table 4. By comparison analysis, the error of the first five natural frequencies obtained by the virtual material method based on Persson contact theory is within 5%. The error of the virtual material method based on the M–B fractal contact theory is within 10%.

## 4. Conclusions

According to the authors in [5], the first five modal shapes obtained by the virtual material method are consistent with those obtained by the experiment. In addition, it is found that the virtual material method can better simulate the metal joint interface than the spring damping method and the finite element method. Therefore, it can be proved that the virtual material method can be equivalent to the metal joint interface. However, the traditional virtual material method is based on fractal contact theory and has certain limitations. The contact mechanics model ignores the interaction between the asperities. When the pressing force (actual contact area) is small, this assumption is approximately true, but when the pressing force (actual contact area) is relatively large, the interaction between the asperities cannot be ignored. Therefore, it will affect the establishment of the equivalent model. In this paper, the Persson contact theory is used to compute the physical properties of virtual materials. The Persson contact theory takes into account the interaction between the asperities. In addition, when the pressing force (actual contact area) of the joint interface is larger, the more accurate the result is, which is suitable for the mechanical joint interface. According to the results of modal experiment and modal simulations, the fact that the error of the first five natural frequencies obtained by the virtual material method and the modal experiment based on Persson contact theory is within 5%, compared with the 10% error of the traditional virtual material method, is a great improvement. Therefore, the physical parameters of virtual materials obtained by Persson contact theory are more accurate, which is more suitable for establishing the virtual material method, so as to achieve accurate analysis of the performance of the whole machine.

## Figures and Tables

**Figure 1 materials-12-03150-f001:**
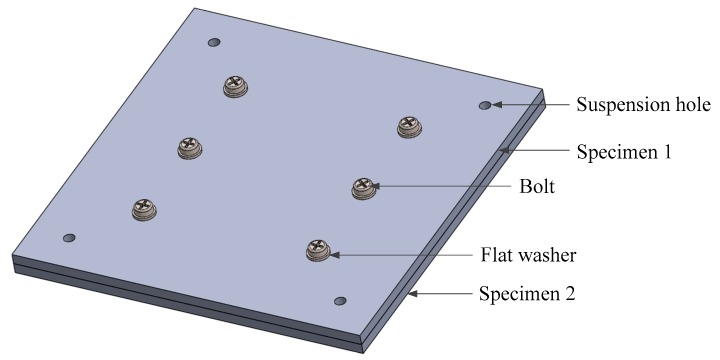
Schematic diagram of the experimental specimens.

**Figure 2 materials-12-03150-f002:**
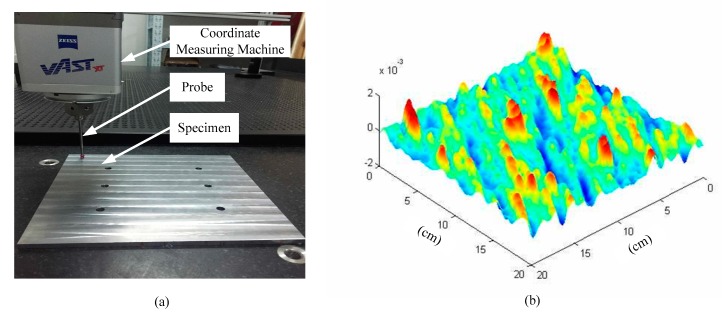
(**a**) acquisition of surface height data, and (**b**) a computer-generated rough substrate.

**Figure 3 materials-12-03150-f003:**
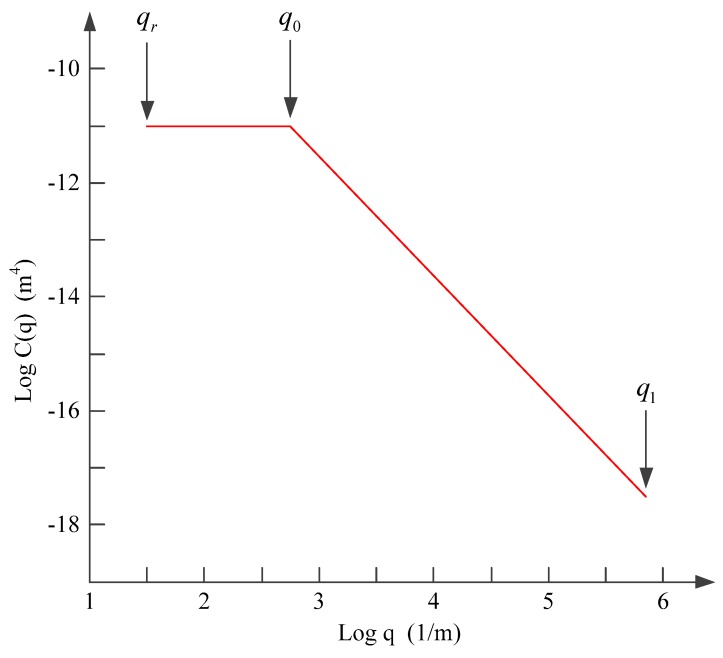
The surface roughness power spectrum.

**Figure 4 materials-12-03150-f004:**
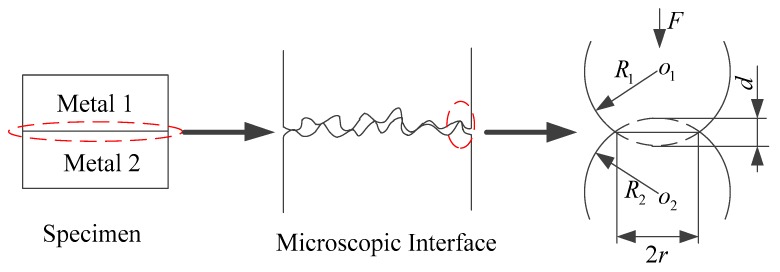
Contact deformation of macroscopic metal joint interface to microscopic contact point.

**Figure 5 materials-12-03150-f005:**
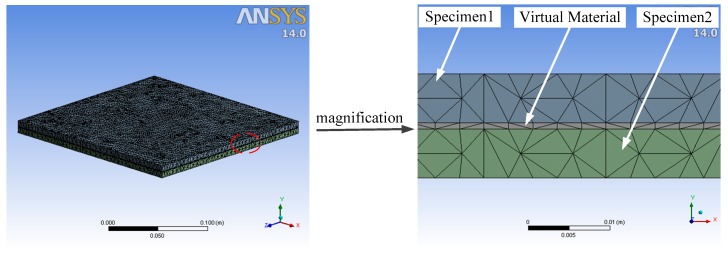
Grid diagram of the virtual material method.

**Figure 6 materials-12-03150-f006:**
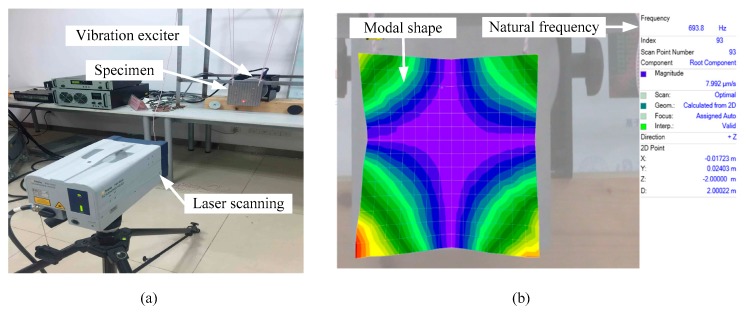
(**a**) experimental setup, and (**b**) modal analysis of the first order by computer.

**Table 1 materials-12-03150-t001:** Physical properties of specimens.

Parameter	Specimens
Elastic modulus: E(GPa)	209
Poisson‘s ratio: μ	0.3
Density: $\rho \left(\mathrm{kg}/m^{3}\right)$	7850
Roughness: Ra(μm)	3.2
Yield strength: $\sigma_{y}\left(\mathrm{MPa}\right)$	355
Hardness: H(MPa)	190

**Table 2 materials-12-03150-t002:** The first five natural frequencies of the model obtained from modal simulation.

Frequency	$f_{1}\left(\mathrm{Hz}\right)$	$f_{2}\left(\mathrm{Hz}\right)$	$f_{3}\left(\mathrm{Hz}\right)$	$f_{4}\left(\mathrm{Hz}\right)$	$f_{5}\left(\mathrm{Hz}\right)$
Persson contact theory	720.4	1141.3	1355.0	1524.1	1635.1
M–B contact theory	745.5	1201.8	1403.8	1600.1	1725.2

**Table 3 materials-12-03150-t003:** The first five natural frequencies of the model obtained from the modal experiment.

Frequency	$f_{1}\left(\mathrm{Hz}\right)$	$f_{2}\left(\mathrm{Hz}\right)$	$f_{3}\left(\mathrm{Hz}\right)$	$f_{4}\left(\mathrm{Hz}\right)$	$f_{5}\left(\mathrm{Hz}\right)$
Modal experiment	693.8	1093.8	1292.2	1459.4	1567.2

**Table 4 materials-12-03150-t004:** The errors of the first five natural frequencies obtained from modal experiment and modal simulation.

Order	1	2	3	4	5
Error of Persson	3.78%	4.34%	4.86%	4.43%	4.34%
Error of M–B	7.45%	9.87%	8.57%	9.63%	10.0%

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
