# Peer review of "Equivalent Method of Joint Interface Based on Persson Contact Theory: Virtual Material Method"

_materials, 2019, doi:10.3390/ma12193150_

Round 1

Reviewer 1 Report

The idea of substituting contact interface with a virtual element layer is innovative. Nevertheless the need of using fractal theory and possible advantages of it is not clear from your paper. Besides some old works dealing with similar problems (Panagouli, Panagiotopoulos) could help.

Reviewer 2 Report

The paper presents an original method of joint interface of solids. The modeling and experimental investigation were conducted. Paper is well-written and structured. 

I have some minor comments:

The specimen's grade is not clear (45# steel). Please add some international grade equivalents. 

Both investigated elements were made of the same material. So it could be used on column (table1) for both materials.

Please add in the 3rd section the version of Solidworks and ANSYS.

Please comment on what is the acceptable surface roughness max-min value to obtain accurate results?

What was the surface finish of the experimental plates? (table 1 relies to the experiment of simulation?)

Please explain why the same grade of materials was investigated? 

Reviewer 3 Report

Dear Authors,

Congratulations on your work, which is very interesting and is very well presented. The paper is excellent, but can be improved in two ways:

The references are used in large groups and don't discribe accurately the previous works carried out in this field. Thus, the Literature Review must be improved. At the end of the results, there is no discussion, crossing other works and results previously performed, comparing the findings achieved now with other obtained in the past. the conclusions seems to contain some discussion about the results.

Congratulations and good luck.

kind regards,

FGS
